# STAM-CCF: Suspicious Tracking Across Multiple Camera Based on Correlation Filters

**DOI:** 10.3390/s19133016

**Published:** 2019-07-09

**Authors:** Ruey-Kai Sheu, Mayuresh Pardeshi, Lun-Chi Chen, Shyan-Ming Yuan

**Affiliations:** 1Department of Computer Science, Tunghai University, Taichung 40704, Taiwan; 2Electrical Engineering and Computer Science Department (EECS-IGP), National Chiao Tung University, Hsinchu 30010, Taiwan; 3Department of Computer Science, National Chiao Tung University, Hsinchu 30010, Taiwan

**Keywords:** suspicious tracking, surveillance, multi-camera tracking, feature based tracking

## Abstract

There is strong demand for real-time suspicious tracking across multiple cameras in intelligent video surveillance for public areas, such as universities, airports and factories. Most criminal events show that the nature of suspicious behavior are carried out by un-known people who try to hide themselves as much as possible. Previous learning-based studies collected a large volume data set to train a learning model to detect humans across multiple cameras but failed to recognize newcomers. There are also several feature-based studies aimed to identify humans within-camera tracking. It would be very difficult for those methods to get necessary feature information in multi-camera scenarios and scenes. It is the purpose of this study to design and implement a suspicious tracking mechanism across multiple cameras based on correlation filters, called suspicious tracking across multiple cameras based on correlation filters (STAM-CCF). By leveraging the geographical information of cameras and YOLO object detection framework, STAM-CCF adjusts human identification and prevents errors caused by information loss in case of object occlusion and overlapping for within-camera tracking cases. STAM-CCF also introduces a camera correlation model and a two-stage gait recognition strategy to deal with problems of re-identification across multiple cameras. Experimental results show that the proposed method performs well with highly acceptable accuracy. The evidences also show that the proposed STAM-CCF method can continuously recognize suspicious behavior within-camera tracking and re-identify it successfully across multiple cameras.

## 1. Introduction

Surveillance for security and monitoring serves the purpose of human or object intrusion detection/trespassers within the protected environment. Traditional approach involves camera based human supported monitoring, which is a challenge to monitor several camera screens at the same time by one or multiple guards specifically within a factory, smart city, research laboratory, university campuses and various commercial and private sectors [1]. There are usually many stationary cameras on the streets or specific areas of a factory. For real cases in Taiwan, there are at least 1000 cameras in a physical crystal optical factory to ensure the safety of dock workers, and detect intrusions. For the same case, twelve dedicated employees of the environmental safety department spent four hours per day to check whether dock workers following the working procedures or not. For another example, it takes average of 1–2 days by uncountable police officers to find out the escaping path of critical crimes such as bank robbing by manually image tracking from cameras standing on the main cross streets.

There are several scenarios are taken into consideration for the development and implementation of suspicious tracking across multiple cameras based on correlation filters (STAM-CCF). For universities, it is usually an open and large area and almost everyone can get into campus without identification checking. For the safety of students, once an intruder gets into the campus and is being identified as a suspicious person, the video surveillance system should automatically track and show the path of the movements of the intruder to the related administrators and security guards. For the use cases of factories, usually the areas and spaces are designed for specific fabrication processes with high-cost equipment. Only authorized persons can get into such working areas. Once a suspicious person breaks in, an alarm will be triggered, and the security system would send the images, movements and location of the intruder to management. The STAM-CCF will play the role of event trigger and information provider in factory scenarios. For future applications such as smart city, a smart city consists of several aspects for its smooth functioning. Traffic management is one of the crucial parts, which needs to be managed. If over speeding is detected of any kind of vehicle, then several accidents can be avoided before time. Route congestion can be notified in advance before heading towards it. Accident cases can be detected on highways, thus alerting nearby hospitals to provide emergency help services. Disaster management is a state emergency that can detect public rush in a particular area affected by flooding, tornado, earthquake, etc., to get rescue help. A disease outburst can also be avoided if suspicious patterns are captured in an area, where multiple people are getting sick and hence can be helpful to be controlled on time. Also, if a suspicious object is kept in a public area such as market, airports, railway stations, parks, it can be a cautious step towards safety. STAM-CCM will play an important role for security surveillance in the future.

### 1.1. STAM-CCF Objectives:

Recently, several surveillance techniques use deep-learning based object detection algorithms that help to count new objects or people count registered. Alternative solutions do exist in the market but they are incapable of operating without a human support. Thus, a need is felt to devise a system that can undertake suspicious tracking without human support. To improve the accuracy of tracking suspicious persons, the proposed correlation filter model involves two major steps: The first is a correlation filter based suspicious tracking with YOLO detector feedback and the second is two-stage gait recognition strategy based across camera tracking method. Eventually, STAM-CCF objectives can be presented as follows:
Solving cold-start problems of tracking suspicious persons: Suspicious tracking is the crucial factor in any surveillance system. The security guard would identify a person or object to be categorized as suspicious, and then try to track the path about suspicious’ movement in a specific area. Generally speaking, several patterns/training needs to be provided to the system and improve its efficiency for long-term residents. For most cases, a suspicious would be a newcomer to a specific monitored area. STAM-CCF aims to recognize such suspicious behavior by using correlation filters and YOLO object detection to reduce the pre-training cost and efficiently solve the cold-start problem of newcomers. YOLO is a popular object detection system in real time.Enabling across multiple camera tracking by features: Multiple objects/humans need to be uniquely identified by features within an image. The proposed model is capable and adaptable to differentiate and track objects or humans by pre-trained YOLO human objects, correlation filtered objects and real-time postures in an image frame. Feature selection is based on vectors of the image parts and used to enable across multiple camera tracking by several ideas from deep learning, possibility and similarity comparisons algorithms.Effectively tracking of suspicious persons on-demand: To ensure the performance and accuracy of suspicious tracking, STAM-CCF leverages entry and exit locations within the protected environment, so that a suspicious person can be tracked across cameras uniquely by a relay. Only adjacent cameras are candidates used for re-identification process, and it will reduce the computation time and cost as well as increasing tracking performance and accuracy in daily operations.


### 1.2. Literature Survey:

Previous within-camera tracking related studies majorly focus on one-camera application scenarios and aggregate all video streams in a grid screen monitored by a human. Due to the inconvenience of usage, time-consuming and human support cost, the research on multi-camera tracking came out with some results recently. Several intelligent video surveillance functions were also recently introduced to the market, and can be divided into two kinds of methods, which are learning-based and non-learning based. The idea of STAM-CCF mainly comes from previous multiple camera related studies, especially for the combination usage of different features-based methods, and the usage of a two-stage mechanism. In addition, for the implementation purpose, several image pre-processing techniques were also introduced in the following.

Suspicious tracking and detection surveys presented by Tripathi et al. [2], have been studied extensively to detect various kinds of abnormal behaviors for images from a single camera. Non-learning-based methods are used to compare fine-grained characteristics to classify user behaviors. For example, feature extraction is performed on an image to capture features either by way of feature based or by tracking based of a particular person. Users can capture feature-based tracking by using several ways of detection by a change in head positions, change in boundary box by a particular height, and width, etc. The classification methods applied include various aspects of person based on posture, shape of the body and motion captured in the video. Several classifiers are included within its experiment and they are k-Nearest Neighbor, Support Vector Mechanism, Hidden Markov Model, Multi-Class Support Vector Mechanism, Haar-feature based cascade classifiers (HAAR), Four-layered Multilayer Perceptron (Four-layered MLP), Fuzzy Logic, etc.

Another widely used one-camera application scenario is the static camera. Such kind of applications are quite progressed through research, regarding theft detection in day-to-day life events in public/private zones. Banks and airports have daily customers and travelers, so human activities are recognized based on single-threaded ontology to avoid robbery and attacks respectively, as presented by Akdemir et al. [3]. Unusual activities/suspicious activities are also detected by a fuzzy c-means algorithm using a ratio histogram suggested by Chuang et al. [4]. Robbery cases are also detected using a forward-backward ratio histogram and a finite state machine (FSM) which proved to have better accuracy and efficiency than that proposed by Chuang et al. [5]. Assaulting and theft/stealing is also detected by using a stochastic representation scheme and a hierarchical algorithm for probabilistic recognition. Automated learning of group activities can be used to overcome the limitation of a system that faces struggles amongst group interactions as presented by Ryoo and Aggarwal [6]. Snatch theft is also one of the growing issues that is overcome by using extracting information from optical flow to detect suspicious activities using surveillance video. In the next stage, to confirm whether the theft actual occurrence is done by using flow pattern statistics is being presented by Ibrahim et al. [7]. Automated teller machines (ATM) crimes are also considered to detect human unusual activities by using multiple object detection and noticing suspicious activities. A classifier is used as utilizing features within the surveillance video. Here, the system may face a limitation of partial occlusion as presented by Sujith [8].

Multi-camera coordination and control (MC3) surveillance strategies provides survey for several state-of-art multi-camera coordination and control strategies presented by Natarajan et al. [1]. There are several computer vision techniques for detecting an object, tracking its movements, analyzing its behavior and storing its pattern as presented by Valera et al. [9]. Techniques developed for activity analysis that are used in surveillance by using trajectory based is presented by Morris et al. [10]. By integrating various sensors, multi-sensors face challenges associated with how the modality is selected, i.e., how can the data fusion be achieved? Furthermore, how are the sensors planned for their implementation? These questions were presented by Abidi et al. [11]. Multi-cameras are also used in object detection and its tracking by using several computer vision techniques as presented by Javed et al. [12]. Hybrid techniques of computer vision, data fusion and smart camera networks explore various limitations in architecture, middleware, calibration and compression in multi cameras as shown by Aghajan et al. [13]. Other computer vision techniques include synchronizing cameras, the communication done within them, and the target correspondence, all of which is presented by Kim et al. [14]. Platforms for hardware and software required for wireless sensor nodes for video surveillance is presented by Seema et al. [15]. Acquiring the highlights from wide area network-based camera technology and its analysis is presented by Chowdhury et al. [16]. Algorithms for the purpose of image processing, vision computing, video coding and visual sensor network platforms is presented by Tavli et al. [17]. Re-identification process used for people re-identification in practical issues and some genuine ways to overcome it is presented by Vezzani et al. [18]. Sparse camera network for capturing target correspondence across cameras and summarizing its activities by using computer vision techniques is presented by Song et al. [19]. A convolution neural network with the loss function similar to neighborhood components analysis (NCA-net) had been proposed to tackle challenges of multiple objects across multiple cameras. Such challenges are caused by low resolution, variation of illumination, complex backgrounds and posture change [20]. Integration of different computer vision and pattern recognition in surveillance done by multiple camera’s using computer vision techniques is presented by Wang et al. [21]. Security and privacy protection for privacy preserving techniques, major security threats and challenges related to visual sensor networks, which are able to perform on board processing and communication components is presented by Winkler et al. [22]. Self-configuration of autonomous smart cameras used for discovery of task topology, calibration, allocation of task and active vision are the various challenges in smart camera network presented by Miguel et al. [23].

There are still many image processing skills used in the STAM-CCF implementation phase to overcome the problem of low-quality images gathering from multiple cameras. These pre-processing skills are important for learning based algorithms whose learning model is built based on good image quality. Robust visual tracking overcomes the challenges faced by non-stationary image streams changing over time in the presence of significant variations of surrounding illumination. Referring to fixed appearance models ignoring shape changes or lightning conditions. Ross et al. [24] presented a tracking method that learns incrementally low-dimensional subspace representation, which displays efficiency in adapting to changes online in the target appearance. There are two important features used of principal component analysis for improving overall tracking performance. These are methods for achieving accuracy in every update of the sample mean and a forgetting factor used for fitting older observations is provided with less modelling power. This work contributes significantly for tracking algorithm in indoor and outdoor environments, including the scaling, posing and illumination changes in the target object. One of the popular techniques, that overcomes the issues of traditional biometrics is the gait recognition by J. Luo et al. [25]. The objectives are achieved by using gait energy image (GEI) and accumulated frame difference energy image (AFDEI), which are gait sequences static and dynamic characteristics, with time characteristics, respectively. Hence, compared to single feature, it obtains better results by combination of above two characteristics. The experimental results claim to have generated better results on the CASIA_B gait database with a high recognition rate. CASIA_B, a gait dataset, was created from the Institute of Automation, Chinese Academy of Sciences.

## 2. Materials and Methods

In this section, we are going to design the planning by deciding the criteria for camera position, within the departmental building and their scanning based on optimal tree traversal to avoid scanning of all cameras for optimal performance.

In Figure 1, the planning for STAM-CCF is done by arranging the positions of cameras within the university campus. Cameras are positioned within certain distances, so that resources are optimized for its usage. Also, the cameras are placed in a specific radius that can cover a particular zone of the campus. The camera numbers are assigned clock-wise and the radius is decided as per the range of the camera. Several entrances are marked within the radius, to gain surveillance of each zone. The above presented plan is quite flexible and can be applied in companies, banks, factories, industrial areas and smart city. In Figure 2, we can analyze it as the same Figure 1 but in a side view rather than top view. The floors shown, not limited to four, each have entry/exit from both the directions. The floor plan can be used within the departmental building of the university campus, company offices, hospitals, smart city, etc. It is easy to track suspicious person in building, in comparison to campus as only two or less cameras are involved on each floor.

In Figure 3, we show how the cameras are scanned for optimal performance achievement. The cameras in the above tree were placed as per the zones, i.e., the first camera zone covers the fifth and sixth camera views, the second camera zone covers the sixth and seventh camera view, and so on. Each camera is placed on an entry point from four directions assuming area coverage of 25 m. Here we use iterative deepening—depth first search [26] algorithm for the tree traversal. The results generated are as follows:
Round 1: 1-2-3-4,Round 2: 1-5-6-2-6-7-3-7-8-4-8-5,Round 3: 1-5-9-12-6-9-10-2-6-9-10-7-10-11-3-7-10-11-8-11-12-4-8-11-12-5-9-12


It can be noticed that in first round we check for all entry level cameras. In round 2, we check as per the zone covering adjacent cameras and in round 3, we cover the first half of the innermost circle from different directions covering each quarter. Hence, this method is found optimal to track suspicious persons within the university campus plan. Whereas, for building floors it is not important, as it only has two entry/exit points on each floor, sequentially.

### 2.1. System Methodology

The proposed methodology, as shown in Figure 4, is categorized into five parts: The input video taken from video surveillance records for suspicious tracking, human tracking and detection, feature extraction, camera relation model/gait sequence and the output for the track map of suspicious person.

Input: As discussed before, in the first part, the input video is selected by the operator of a specific date and the suspicious person to be tracked is selected.

Human Tracking: In the second part, human detection and tracking process is initialized. Here, the person is detected using YOLOv3 [27] and his bounding box is estimated to generate four coordinates t_x_, t_y_, t_w_ and t_h_. For the image calculation, the cell is taken from the top left corner as c_x_, c_y_ and p_w_, p_h_ is the width and height for the bounding box. Hence the prediction is detailed as:
(1)$$b_{x}=\sigma (t_{x})+c_{x}$$
(2)$$b_{y}=\sigma (t_{y})+c_{y}$$
(3)$$b_{w}=p_{w}e^{t_{w}}$$
(4)$$b_{h}=p_{h}e^{t_{h}}$$


The detection is then continued with a correlation tracker, which is then overlapped on the same selected person. Here, we take 2D Fourier transform as input image F = F (f) and of the filter H = F (h) are computed as correlation in Fourier domain fast Fourier transform (FFT).
(5)$$H*=\frac{\sum_{i}\left(G_{i}⊙F_{i}^{*}\right)}{\sum_{i}\left(G_{i}⊙F_{i}^{*}\right)}$$


The definition of the H filter, also known as minimum output sum of squared error filter (MOSSE) [28], is shown in Equation (5). Fi stands for training images. Gi is the training output. ⊙ denotes for element-wise multiplication and * represents the complex conjugate. The two filters are then taken as intersection over union of box as specified in later Equation (6).

### 2.2. Detection and Tracking Model

As shown in Figure 5, the human tracking flowchart (a), a recorded video is selected from a particular day, which is supposed to be inspected. The selected video is then played to see a pedestrian entry within the campus, then the operator chooses a suspicious person/pedestrian by mouse pointer, to be tracked. The selected pedestrian is then highlighted with a box by using YOLO [27] detection and thus tracker on that the pedestrian is initialized using a correlation filter (CF) tracker. The tracker helps to highlight a moving pedestrian in successive frames. Next the intersection over union (IoU) checker [29] is activated that helps to take the average of intersected box over the union box for the accuracy of the box. The IoU for achieving similarity between the boxes A, B ⊆ S ∈ R^n^ is given by
(6)$$\mathrm{IoU}=\frac{\left|A\cap B\right|}{\left|A\cup B\right|}$$


The details of the IoU checker is given in Figure 5b. The IoU checks by YOLO prediction over CF tracker and calculates the IoU. If the calculated IoU is greater than the threshold, then update the tracker of YOLO, else continue to generate output.

The system flowchart shown in Figure 4 only represents detection and tracking. In the next part, a detail representation of the system methodology diagram will be discussed including feature extraction and camera relation model to get an output from the system.

Feature Extraction: After detection and tracking, we move towards the third part of feature extraction. In feature extraction, we are doing pose estimation by selecting 14 of 18 pose features using open software [30]. After completing pose estimation, we provide input to Gait to generate the Gait sequence and to identify the person uniquely [25]. The gait sequence is generated for every person, for each twenty to thirty frame sequence depending on the parameter setting, as it also affects the feature dimension for model training. In this model, we have selected 27 × (14 × 2) dimension as 27 frames as a cycle, 14 feature points and 2 as (*x*, *y*) axis of the image. To calculate gait energy image of the selected person, which is weighted average method for gait sequence of cycles, we need to obtain *G*(*x*, *y*)
(7)$$G\left(x+y\right)=\frac{1}{N}\sum_{t=0}^{N}B\left(x,y,t\right)$$


Here, *N* refers to number of frames in sequence of gait cycle, t is the number of gait frames and *B*(*x*, *y*, *t*) is the gait cycle image sequence. We then calculate the accumulated frame difference image, which is given as *A*(*x*, *y*)
(8)$$A\left(x,y\right)=\frac{1}{N}\sum_{t=1}^{N}F\left(x,y,t\right)$$
where, *F*(*x*, *y*, *t*) are used to represent backward frame difference image. The accumulated frame difference image is used to indicate time characteristics of human walking, so as to optimize the amount of computation. The feature extraction is then performed by using *f*(*x*, *y*) for 2D gray image, its origin moment and central moment both given as (*p* + *q*)^*th*^ separately.
(9)$$m_{pq}=\sum_{x}\;\sum_{y}x_{p}\;y_{q}\;f\left(x,y\right),\;p,q=0,1,2\ldots .$$
(10)$${µ}_{pq}=\sum_{x}\;\sum_{y}{\left(x-\bar{x}\right)}^{p}{\left(y-\bar{y}\right)}^{q}f\left(x,y\right),\;p,q=0,1,2\ldots .$$


Here, $m_{pq}$ and ${µ}_{pq}$ are the origin and central moment of the (*p* + *q*)^*th*^, whereas the centroid coordinates of image are marked by $\left(\bar{x},\bar{y}\right)$, as $\bar{x}$ = m10/m00 and $\bar{y}$ = m01/m00. Later on, we normalize the central moment (*p* + *q*)^*th*^ by
(11)$$\eta_{pq}=\frac{{µ}_{pq}}{{µ}_{00}^{r}}$$
where *r* = (*p* + *q* + 2)/2. Also, when we have images to be compared from different cameras then a possibility may be the different pixel image comparison of the same suspicious person as a requirement, i.e., 32 pixels and 128 pixels. We use normalization of suspicious person features, so as to compare them efficiently. It is also known for calculating proportional distance between nodes.

Camera Relation Model/Gait Recognition: In this part, we can track the user trajectories, i.e., when a suspicious person enters/exits a particular area or wears different clothes or a hat. For the remaining part of the two-stage gait recognition, which never misses a suspicious person, details can be seen in Section 2.4. Two-stage strategy for Gait recognition.

Output: The camera relation model provides us with the track of the selected suspicious person and the two-stage gait recognition helps us to keep track across multiple cameras. Ultimately, we receive the complete track of the suspicious person using an adaptive model.

### 2.3. Suspicious Person Tracking

As shown in Figure 6, the objective of this diagram is to explain specifically how the selected suspicious person is detected and tracked within the system. As soon as we select the suspicious person to be tracked within a video, immediately starting from the first frame correlation filter tracker [28] detecting the person to be tracked by the bounding the box and at the same time maintaining for accuracy. Here, the bounding box consists of x, y coordinates of the upper-left corner, width and height. The bounding box calculated in last frame and current frame image are used as input information of the correlation filter tracker. Then we detect the same person with YOLOv3 [27] by a different bounding box overlapping to each other. In the next frame, the IoU matrix is calculated as specified in Equation (1), which is intersection over union of bounding boxes of correlation filter tracker and YOLOv3. Henceforth, this IoU matrix [29] technique provides more accuracy in suspicious person identification and detection. If the calculated matrix value exceeds a threshold then it will update the tracker by the YOLOv3 bounding box later on, as shown in Algorithm 1, or it will be recalculated for the correlation filter for tracking the suspicious person.

### 2.4. Gait Recognition Using long short term memory (LSTM)

The long short-term memory (LSTM) network shown in Figure 7, is used to detect a suspicious person accurately from different angles or poses. For gait based human identification, the skeletal coordinates of 14 posture points are extracted as feature points using OpenPose [30], and then 27 continuous poses are selected as a gait sequence. There are total of 27 LSTM timestamps that are executed to identify a suspicious person from different gait sequences. Figure 7 shows a 27-timestamp based LSTM network representing a gait sequence (gait cycle). Each LSTM function is representing seven moment invariants which are calculated from 14 feature points by hu moment for translation, scale, rotation and reflection. The unit=100 is used to represent the number of hidden layers for each LSTM.

As there is no training, weights are being assigned randomly in LSTM and to extract features, use the same model. The extracted features help to calculate the distance between two gait sequences. The number of classifications of images for identification, which are then taken as average and classified for matching for 10 percent times distance calculation thus reducing the noise. For nearest neighbor distance calculation average distance is useful. At last we pass the output as input to softmax function [31] for calculating the multivariate classification probability:
(12)$$f\left(x_{i}\right)=\frac{\exp\left(x_{i}\right)}{\sum_{j}\exp\left(x_{j}\right)}$$


In Equation (12), *i*, *j* = 1..*k* ϵ $R^{K}$. The exponential function is applied to each element of $x_{i}$ as the input vector and by dividing sum of all exponentials we get normalized values within the interval (0, 1).

### 2.5. Two-stage Strategy for Gait Recognition

In Figure 8, long short-term memory (LSTM) is used in conjunction with a Gait energy image [32] to have features learned with temporal information and cross view gait recognition. The gait information of this estimation method is stored in one frame by using heatmaps given as input to convolutional neural network. The distance between sequences can be calculated as
(13)$$DIST\left(P,\;G\right)=\sum_{i=1}^{n_{p}}\min_{j}d\left(p_{i},g_{j}\right)$$


In Equation (13), let the feature vector extracted from a gait sequence in probe set and gallery set be given as: *P* = {*pi* |*i* = 1, …, *np*} and *G* = {*gi* |*i* = 1, …, *ng*} respectively. Here, d($p_{i},g_{j}$) is used to denote cosine distance. As shown in Figure 8, when the suspicious person is crossing the camera area, the next camera is used to detect the same. Otherwise, in case the person is not identified then we give input to K-nearest neighbors (KNN) for re-identification by the gait sequence. KNN used for classification accuracy with better achieved speed by using good value for k = 5 [33]. Figure 8 shows when multiple persons are in CAM1 are labelled as A, B and C then while appearing in CAM2, they are again labelled as A, B and C depending on their feature points accurately. So even if miss by LSTM is occurred, it is then passed to KNN for labelling and continued tracking.

### 2.6. Cross Camera Metadata

As shown in Figure 9 and Figure 10, camera map and metadata are presented in an xml file format respectively. It is used to represent connection between various camera connected on a building’s single floor. Two pathways are connected in the angle form, so that their ends are the connection areas to each other. As camera 1 and camera 2 have a single connecting point, we can see CAM1 embeds CAM2 within it. Later on, CAM2 embeds CAM1 and CAM3 as CAM2 has a connecting point in between of them. Henceforth, CAM3 is having only CAM2 embedded within it. This embedding helps the suspicious person tracking, when he travels through the path.

### 2.7. Algorithms

As shown in human tracking algorithm of Table 1, we are tracking suspicious persons/humans within a single camera path range. In step 1, the input given to this algorithm is the frame from a single camera. Each frame_i_ is processed by the algorithm. Whereas, in step 2 is the bbox of a selected person identified by the operator is to be tracked. In step 3, the output, an appropriate pedestrian bounding box is given. In step 4, we initialize threshold to be set for IoU between YOLO and the CF box tracker. In step 5, we initialize correlation tracker bbox for correlation tracking. In step 6, the pathset set to null, used to represent the suspicious person’s path travel from entry to exit of the camera area. In step 7, a For loop is set to process each frame_i_ generated by the camera. In step 8, YOLO box will appear on the selected suspicious person on the frame_i_. In step 9, correlation box will appear on the same human and on the same frame_i_. In step 10, threshold of IoU will be calculated between YOLO box and correlation box, so as to get a bounding box with accuracy step 11, the calculated threshold of IOU is checked with the threshold limit set by the operator. In step 12 and 13, if the threshold limit is exceeded by current calculation of the bbox then the path is updated to the tracker for the movement update and hence appended to the path set by the YOLO box. Later on, the pathset is used to evaluate the trajectory of a suspicious person’s path travelled. In step 14, else if the threshold calculate is not exceeding a limit then in step 15, the pathset is appended by the CF box. Step 16 and 17 are used to make the end part of threshold of IoU condition and frame within the particular camera loop respectively. In step 18, a final pathset is returned as the output by the algorithm.

In Table 2, we are capturing the suspicious person’s movement across the camera. In step 1, input is taken by the algorithm from frame_i_ of initial j-th camera, where the suspicious person is selected by the operator. In step 2, k-th camera is selected based on the camera metadata for the connecting camera to be selected. In step 3, the bbox is the selection of suspicious person used for tracking. In Step 4, it shows the output format of algorithm as a suspicious person bounding box set. In step 5, tracking is started by the C_j_ initial camera for the suspicious person. In step 6, the access area is searched from the camera metadata, so that continuous trajectory tracking of a suspicious person is maintained. In step 7, when the suspicious person is found missing from the initial camera C_j_ an event is triggered to continue that person’s tracking by passing bbox and current access area.

Step 8, shows the call to the function on an event made by algorithm when the suspicious person is disappearing. It takes input bbox calculated for the person and the access area that are connecting to that camera by camera metadata. In step 9, C_k_ is the new camera area where the new person is supposed to be tracked. In step 10, once the camera C_k_ is decided then the algorithm access the frames that will be provided by that camera C_k_. In step 11, the suspicious person selected before is identified by the two-stage gait recognition, as per the Section 2.4. Then there is the two-stage strategy for gait recognition. At the end step 12, the bbox tracking is continued for the suspicious person.

## 3. Results

In this section, we are going to present several experiments that are conducted based on methodology of our proposed paper. For conducting experiments, the following system configuration was used, as given below in Table 3. Table 4 gives information regarding camera configuration used for experiment within this paper.

### 3.1. Dataset

As shown in Figure 11, the dataset considered in this paper was recorded using a camera network in the entrance area of the department building. There were three cameras used: camera 1, camera 2 and camera 3, and each of them were covering different paths and had non-overlapping views. In this experiment, we have considered three different scenes where camera 1 and camera 2 are in the corridor and camera 3 is within classroom. The scene shows two students walking together. Whereas during the test scenario, we have included same students with different clothes.

In Figure 12, a different scenario was considered for experiment where dataset is taken from Camera 4, 5 within the cloud innovation school (CIS) and Camera 8 at its surrounding area. Here, Camera 4 and 5 were used for recording the training data of the experiment and Camera 8 was used for performing test that includes four students and the situation of occlusion effect.

### 3.2. Experiments

In Figure 13, we have considered a laboratory environment, in which first we are going checked the YOLO bounding box detection capability. In Figure 13a, it can be observed that YOLO detection method is fast and accurate. Also, it was found to be better than other detection methods but not necessarily it detects every frame as when there will be flashing effects. Due to some limitations, it was noticed that YOLO is not able to track all objects successfully. In Figure 13b, by applying the correlation filter, we were able to detect all of the students/objects, which were missed by YOLO. It was the case that the observed CF was better in tracking objects, even when not trained. The frame became smaller and smaller as the object approached, which may have led to a loss of object in some cases and hence we needed to overcome such a limitation. Also, it led to a loss of object, when encountering an obstacle.

In Figure 14a, we have taken a different scenario, where only a CF was used to track students/objects within the laboratory environment from algorithm 1. Here, it was noticed that there is insufficient training for the CF filter to avoid obstacles that may lead to loss of robustness. Also, the filter suffered from the tracking wrong/error object, like when the object was nearby an obstacle, i.e., the table/desk and when two-person body view was overlapped. It was also noticed that the CF lost the target, when severe changes in object size occured during movement. Technically, while tracking CF may have missed the target because of too much shadowing or updating the wrong template because of the scale relationship. Several such reasons lead to the devising of new method, so as to achieve better results. In Figure 14b, we have combined YOLO and correlation filtering to overcome independent limitations. As we can see from the results, the objects which were missed before due to obstacles/other objects overlapping is now being recovered. For combining YOLO + CF, it helps us to track objects appropriately without any error. Also, even if two objects are overlapping each other, they are separated by different bounding boxes and categorized separately. By using YOLO + CF, it becomes easy to overcome the obstacles and continue tracking of suspicious person. Therefore, the target frame is continuous and stable.

In Figure 15a, as stated before, correlation filter (CF) was not able to overcome the limitation of object size detection and tracking. Therefore, if the object is in continuous movement, the bounding box is found to be lagging the path as shown in the tracking results. To overcome this situation, YOLO + CF was used as shown in Figure 15b. It can be easily noticed here that the object which was lacking the track before by the bounding box is now completely aligned and is able to properly detect and perform its tracking with accuracy. Hence the limitation of size stability was resolved.

In Figure 16, a new scenario is considered for performing detection and tracking across camera. As shown in Figure 16a, a path view is taken from CAM1 as explained before in Figure 9 model. Algorithm 2 was used to recognize the gait features of all people and label them uniquely as the feature detection done using gait. Once the features were detected as shown in Figure 16b and subsequent images, then the algorithm started tracking the detected people. Even in case these people change positions, their subsequent identity and tracking was maintained.

In Figure 17, we considered a new scenario using Figure 9 model for across multiple camera view recognition and tracking. By using algorithm 2, when the features are recognized and tracked in CAM1 in Figure 16, then it is continued to be tracked in CAM2 as given in Figure 17a, fetching gait features for uniquely identifying a person by his postures even if he changes clothes and hat. In Figure 17b, once the fetching of gait features is done then the algorithm identifies the identity assigned to each person before and then continues to track them in the subsequent images. Henceforth, this across multiple camera view tracking allows us to generate tracking of suspicious person within the Tunghai university campus and generates a map at the end for suspicious person detection, recognizing and tracking path. Thus, it will solve the incident, if any, that occurred. Leading to precautionary conditions to be maintained to avoid any harm or damage to be caused in the future, within the campus environment for safety concerns.

### 3.3. Statistics

#### 3.3.1. Experiments on Our Database

The Table 5 represents the value for IoU that computes the area of overlap between the predicted bounding box and the ground truth bounding box. In this experiment, the IoU threshold more than 25% is considered to be true positive.

In Figure 18a, the mean average precision (mAP) is used for the calculating the accuracy of object detection. It is considered to have several other parameters of TP, TP, FP and FN for the calculation for precision value of recall value over 0 or 1. It can be represented mathematically as:
(14)$$mAP=\frac{\sum_{q=1}^{Q}AvgPrecision\left(q\right)}{Q}$$
where *q* is denoted as the number of queries. For YOLOv3 the mean average precision is given as 0.9188. In Figure 18b, the learning curve is evaluated with 200 epochs for gait classification using LSTM model. These statistics is used to represent accuracy of various methods used within methodology section.

The target scenario of pedestrian tracking across cameras is that human objects can be recognized and tracked when they appear across cameras. To assess the efficacy of the proposed method, the precision can be defined as the ratio of correctly recognized when people in different cameras. For cross-camera scenarios in our test environment, the mean average precision is measured as 0.85 with 27 frames as a gait cycle on our dataset.

#### 3.3.2. Experiments for Multiple Object Tracking Benchmark

There are several famous databases, such as Duke [34] and MARS [35], to be used for the evaluation of multiple object tracking algorithms in batch mode. This means that human objects should be collected and trained in advance. For suspicious detection and tracking purposes, it is not possible to collect and train the image data before a criminal event happened. A MOTChallenge [36] database is used for the performance evaluation of STAM-CCF. The MOTChallenge is the leading assessment platform to evaluate the performance of multiple object tracking algorithms. The MOT15 of MOTChallenge database was used as the benchmark database in our experiment for single camera tracking test cases. It consists of 11 datasets with a total of 11286 frames or 996 s of video. For performance evaluation and judgement for online object tracking algorithms, nine evaluation indicators of MOT15 are measured as defined as follows [37,38]:
(1)MOTA: Multi-object tracking accuracy.(2)MOTP: Multi-object tracking precision.(3)FAF: Number of false alarms per frame.(4)MT: Ratio of mostly tracked trajectories.(5)ML: Ratio of mostly lost trajectories.(6)FP: Number of false detections.(7)FN: Number of missed detections.(8)ID Switch: Number of identity switches. i.e., ID switch ratio = #ID switch/recall(9)Frag: Number of track fragmentations caused by miss detection where a track is interrupted by miss detection.


Due to the goal of STAM-CCF is to recognize the suspicious in real time. Several popular online tracking approaches are selected for the comparison of human objects detection and tracking. The performance data of SORT is referred from [39], which are popularly used on the market from 2016, and other methods, including TDAM [40], MDP [41], RMOT [42], TC_ODAL [43], are selected from MOT15 official web site. The results and comparisons are shown in Table 6. STAM-CCF performs outstanding in MOTA, MT, ML and Frag indicators. It also works well for other indicators.

#### 3.3.3. Experiment on CamNeT Database

CamNet is a non-overlapping camera network tracking dataset (CamNeT) for evaluating multi-target tracking algorithms [44]. The dataset is composed of five to eight cameras covering both indoor and outdoor scenes at a university. In contrast to our dataset, the CamNet dataset is used for the evaluation of STAM-CCF for low resolution image frames. It has more than 1600 frames, with a resolution of 640 × 480 pixels, 20–30 frames per second, and more than 25 identities, including five to eight cameras. The dataset has six scenarios, and each scenario video lasts at least five minutes. Due to the low quality of image frames, in contrast to our dataset which can capture 14 posture points per image frame, only five feature points can be used for re-identification estimation. Table 7 shows the precision results of some experiments on CamNet. The average precision is around 60% and it demonstrates that STAM-CCF still works well in low-resolution cameras.

## 4. Discussion

### 4.1. Experience Sharing for STAM-CCF Design

Object tracking applications usually adapt possibility-based algorithms such as Kalman filter, correlation filter or combination or variances of these two methods. Kalman filter use previous object movement tracks to predict the possibilities of next object locations. However, it often fails in cases of missing detection by occlusions, human overlapping, variant illumination and the likes. On the other hand, correlation filter method tries to compare the most similar bounding boxes between continuous image frames. Theoretically, the Kalman filter is suitable for simple use cases of radar-based object tracking where images are consisted of target object and noise signals. To meet the requirement of pre-condition for using Kalman, the image pre-process would cost much to get high-quality images from cameras with different kinds of scenes and scenarios. In addition, from the offline testing results, the correlation filter performs much better than Kalman filter [45,46]. Those are main reasons why STAM-CCF choose correlation filter, instead of Kalman filter, as a base line for suspicious tracking system design within a camera.

Although the correlation filter performs better than Kalman in across cameras tracking scenarios, it also faces problems of missing detection. It will cause the failure of suspicious tracking through many image frames and identify the suspicious back in lucky conditions. From our offline experiments, most of the cases, it mismatches the human objects, and fails to track suspicious persons in such kind of missing detection conditions. This means that it needs additional complementary functions to solve the missing detection problem. Luckily, it turns out the idea of leveraging YOLO object detection functions to get the nearest bounding box with maximum response.

After overcoming within-camera tracking problems, STAM-CCF remains the problem of re-identification of suspicious across multiple cameras. That is, STAM-CCF has to compare all the first human image frames for each camera to identify which human object is the tracking suspicious. Intuitively, only adjacent cameras geographically could be the candidate camera because human movement speed is limited to 36 KM per hour currently. Hence, the related location information of each camera is used to compute the possibility of suspicious and reduce the computation resources and cost very much. Finally, to re-identify suspicious from candidate cameras, the posture and gait features are used to compute the similarity the first images with YOLO bounding box.

### 4.2. Implementation Issues

There are two major procedures of STAM-CCF implementation. The first one is to track suspicious within a camera, and the other one is to compute the similarity between first image of YOLO bounding box between cameras. For the case of within-camera tracking, correlation filter will always suggest the bounding box with maximum response. To ensure the accuracy of the object tracking, the YOLO bounding box of largest IoU value was used to adjust correlation filter bounding box. The assumption was that human has limitation of movement speed and cannot move far than a specific distance between two or five image time frames by using camera of at least 15 fps. Based on the rule, the IoU larger than 0.4 is the threshold, and YOLO bounding box is used to adjust the correlation filter bounding box as the base target. Recursively, the suspicious will be tracked correctly within a camera. Even the experimental results show that STAM-CCF performs well in most use cases, there are still limitations or exceptions should be handled within a camera. Take the overlapping case as an example, the suspicious is correctly identified in the previous image frame but two humans overlap. If the suspicious person is hiding in the background, and another human is in the front ground then there will be only one YOLO bounding box with the largest IoU value. STAM-CCF will fail to recognize the suspicious person at that moment. The implementation should handle such kind of exceptions by counting and keeping the number of YOLO bounding boxes if no human object standing near the boundary of image frames.

The other major procedure is to compute the similarity of posture and gait features for YOLO bounding boxes. Due to the property of the cold start problem, STAM-CCF will not have a large enough data set of suspicious postures and gait features and, thus, might lead into error-prone conditions. For example, a human will stand with different angles facing to the camera. To solve this problem, with the help of location information of camera, only humans in the connection area will be taken as candidates used for similarity comparison. STAM-CCF will response the bounding box with the highest gait feature similarity value.

## 5. Conclusions

Suspicious tracking across multiple cameras has strong demands for application scenarios such as safety assurance of human, intruder detection and alarm, criminal tracking and the like. To enable the multiple camera tracking capability and overcome the obstacles which mainly come with cold-start problems and low-quality image issues, the proposed STAM-CCF combines correlation filters, YOLO object detection, posture and gait features to realize the across camera tracking functions. In addition, it also leverages the location information to ensure good system performance by only re-identifying adjacent or candidate camera images. Besides the sharing of implementation experiences, several scenarios in a university are also designed for STAM-CCF feasibility and performance testing. Due to the limitation of resources, and a tight schedule, there are still different kinds of application scenarios such as factories and smart cities to be deployed and test in the near future. Eventually, the experimental results will show that STAM-CCF performs well with highly acceptable accuracy and performance.

## Figures and Tables

**Figure 1 sensors-19-03016-f001:**
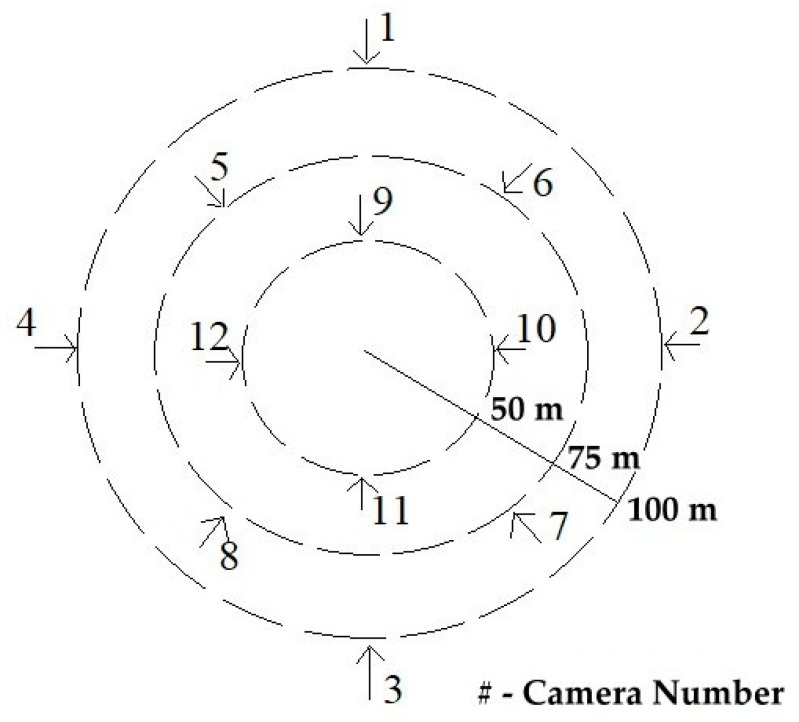
Camera Placements in University Campus.

**Figure 2 sensors-19-03016-f002:**
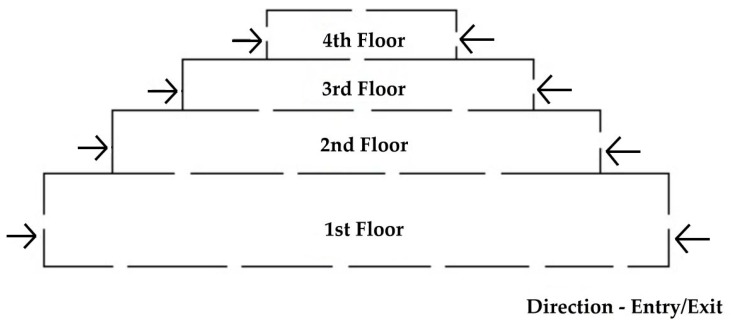
Camera Placements on Building Floors.

**Figure 3 sensors-19-03016-f003:**
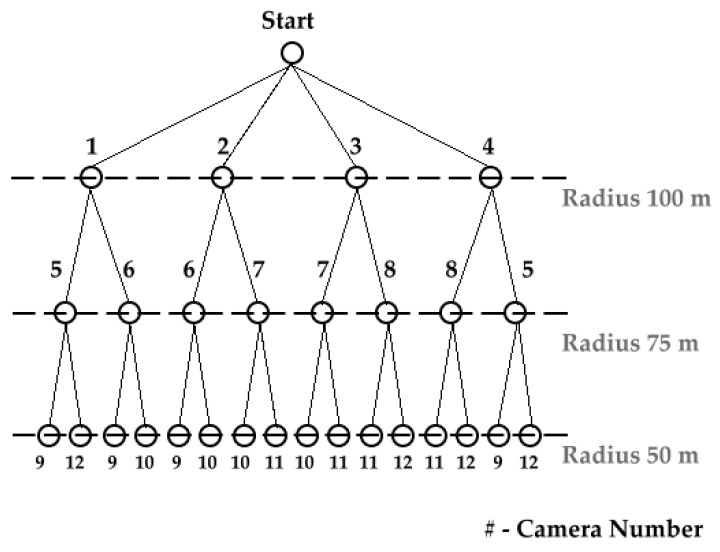
Scanning of Cameras using Tree Traversal Method.

**Figure 4 sensors-19-03016-f004:**
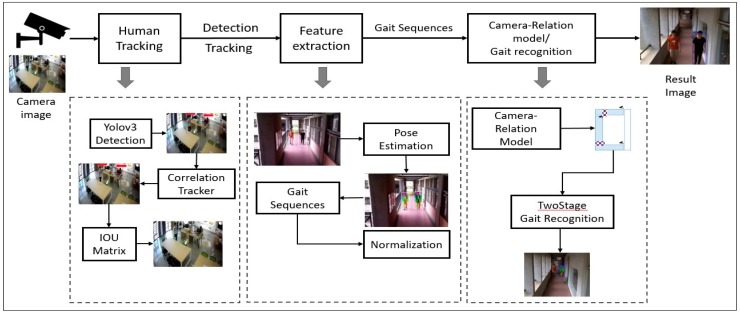
System Methodology

**Figure 5 sensors-19-03016-f005:**
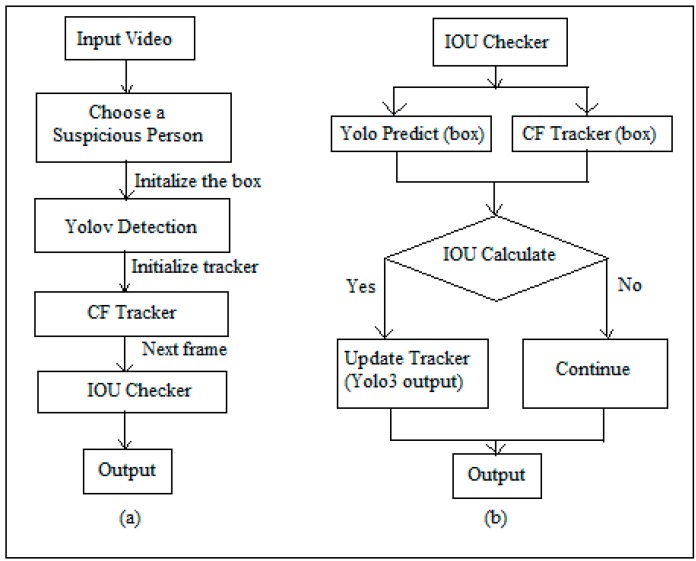
Flowcharts for (**a**) Human Tracking using YOLO predict and CF Tracker, (**b**) IoU Checker.

**Figure 6 sensors-19-03016-f006:**
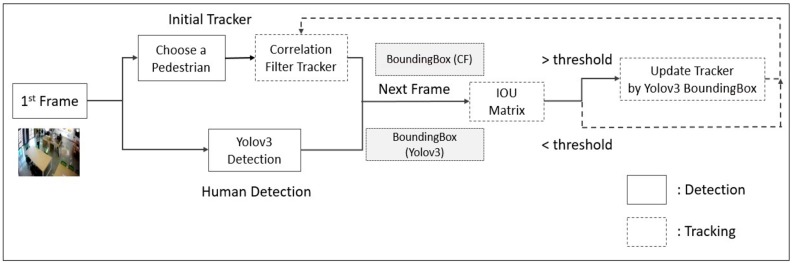
Suspicious Tracking Model.

**Figure 7 sensors-19-03016-f007:**
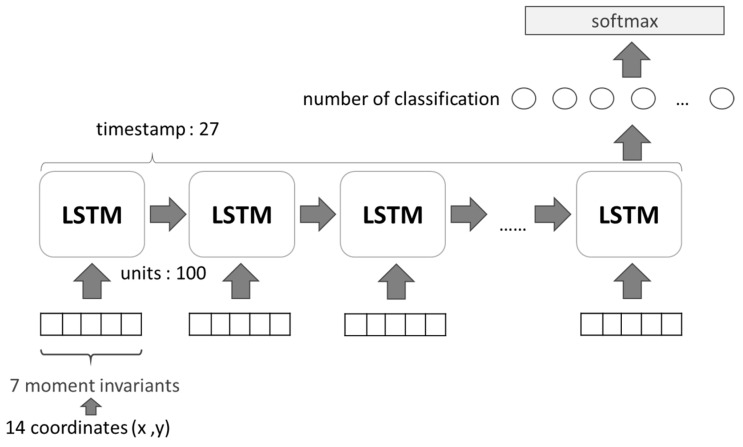
Gait Recognition Using a Long Short Term Memory (LSTM) Network.

**Figure 8 sensors-19-03016-f008:**
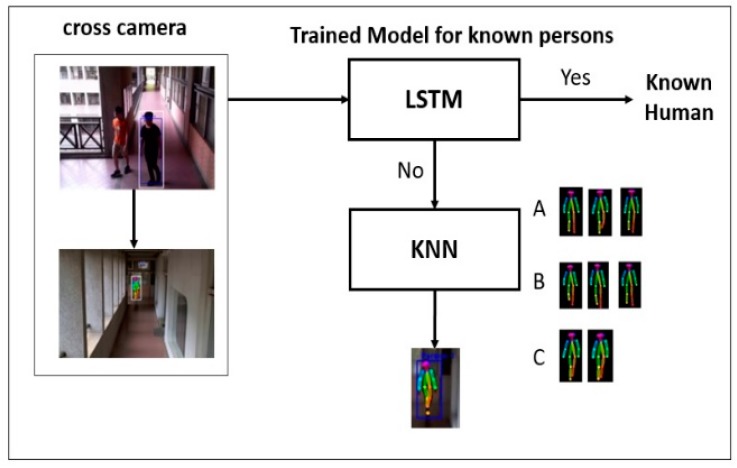
Two-Stage Gait Recognition.

**Figure 9 sensors-19-03016-f009:**
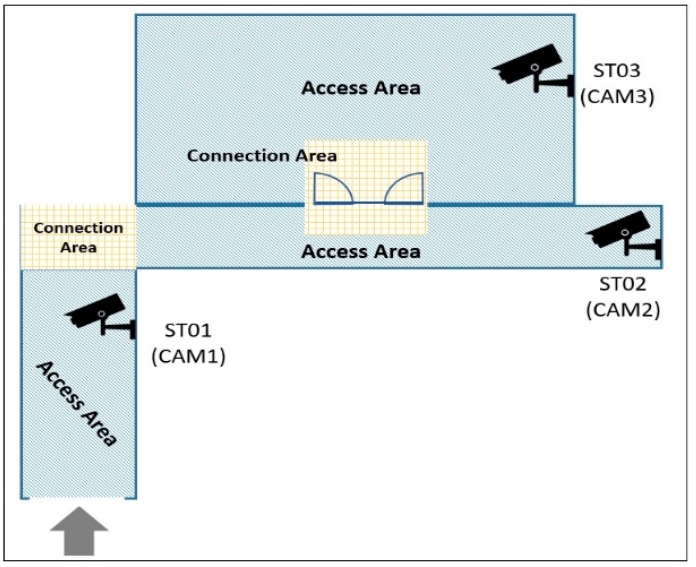
Placement of Cameras within the First Floor of Department Building in University Campus.

**Figure 10 sensors-19-03016-f010:**
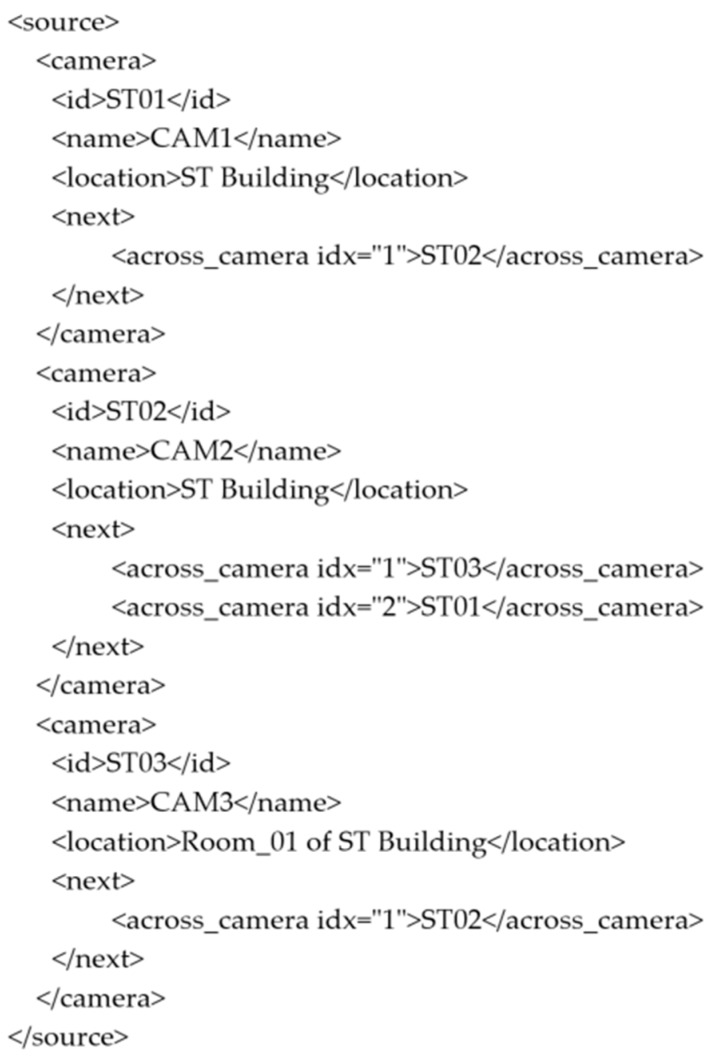
Camera Metadata.

**Figure 11 sensors-19-03016-f011:**
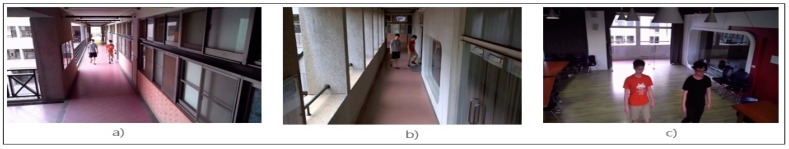
Dataset Taken from Camera: (**a**) Camera 1, (**b**) Camera 2 and (**c**) Camera 3 of Department Building.

**Figure 12 sensors-19-03016-f012:**
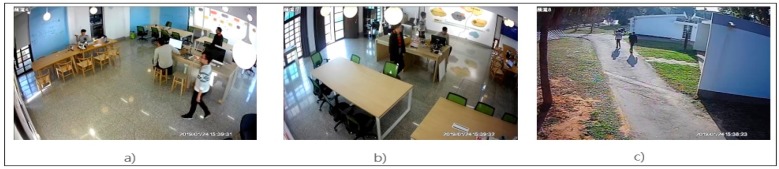
Dataset Taken from Camera (**a**) Camera 4, (**b**) Camera 5 and (**c**) Camera 8 of Cloud Innovation School (CIS).

**Figure 13 sensors-19-03016-f013:**
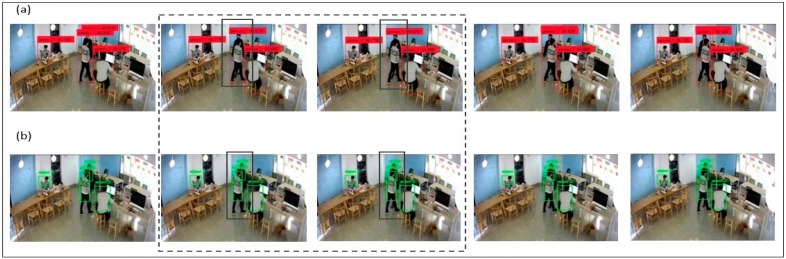
Experiment for (**a**) YOLO Detection of Human with Red Bounding Box, (**b**) Correlation Filtering Detection of Human with Green Bounding Box.

**Figure 14 sensors-19-03016-f014:**
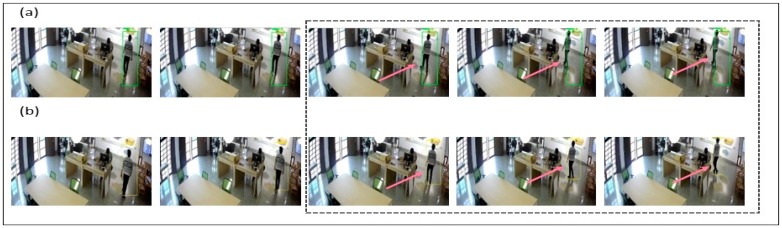
Experiment for (**a**) correlation filter (CF) Detection of Human with Green Bounding Box, (**b**) CF + YOLO Detection of Human with Red Bounding Box.

**Figure 15 sensors-19-03016-f015:**
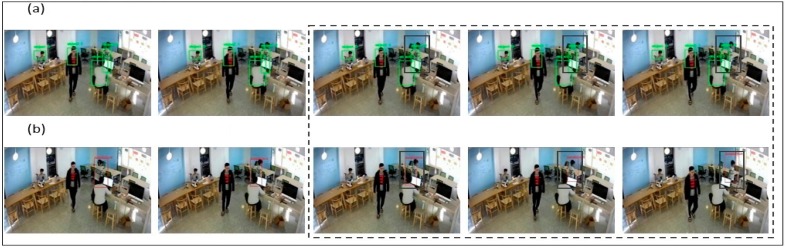
Experiment for (**a**) CF Detection of Human with Green Bounding Box, (**b**) YOLO + CF Detection of Human with Yellow Bounding Box for Size Stability.

**Figure 16 sensors-19-03016-f016:**
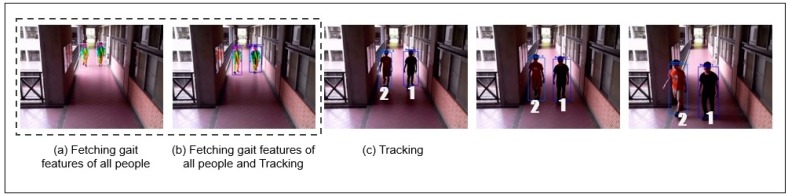
Experiment for Gait Features Detection and Tracking using a Single Camera View.

**Figure 17 sensors-19-03016-f017:**
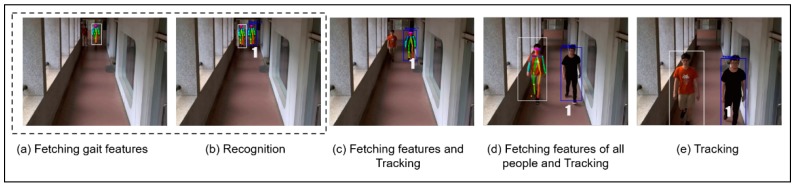
Experiment for Recognition and Tracking Across Multiple Camera View.

**Figure 18 sensors-19-03016-f018:**
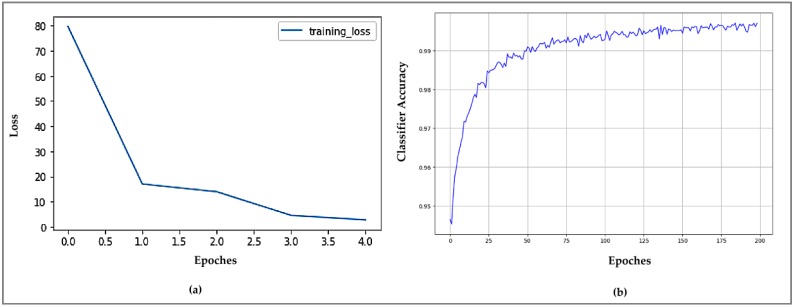
Statistics for (**a**) YOLO Learning Curve for Human Detection and (**b**) Accuracy of Gait Recognition using LSTM.

**Table 1 sensors-19-03016-t001:** An algorithm of Human Tracking using a Single Camera View.

Algorithm 1: Human Tracking Using a Single Camera View
1: **INPUT**: *frame_i_* ϵ *C*, camera frame set
2: *bbox*, a pedestrian bounding box
3: **OUTPUT**: human bounding box set(X_min_,Y_min_,X_max_,Y_max_)
4: *threshold_iou_* = a threshold of IoU between YOLO and CF
5: *tracker* = initialize CorrelationTracker(*bbox*)
6: *pathset* = ф
7: For each *frame_i_* in *C*
8: *box_YOLO_* = humanDetectByYOLO(*frame_i_*)
9: *box_CF_* = humanTrackByCF(*tracker*, *frame_i_*)
10: *t_IoU_* = calculateIoU(*box_YOLO_*, *box_CF_*)
11: If *t_IoU_* > threshold_IoU_
12: *tracker* = updateCFTracker(*tracker*, *box_YOLO_*)
13: append(*pathset, box_Yolo_*)
14: else
15: append(*pathset, box_CF_*)
16: end
17: end
18: return *pathset*

**Table 2 sensors-19-03016-t002:** An Algorithm of Human Tracking Across Multiple Camera View.

Algorithm 2: Human Tracking Across Multiple Camera View
1: **INPUT**: *frame_i_* ϵ *C_j_*, camera frame set of j-th camera
2: *C_k_* ϵ C, k-th camera of camera set C
3: *bbox*, a pedestrian bounding box
4: **OUTPUT**: human bounding boxset(*X_min_*, *Y_min_*, *X_max_*, *Y_max_*)
5: *C_j_* = a pedestrian shown in j-th camera
6: *access_area* = getAccessAreaFromCameraMetadata(*C_j_*)
7: *update_area* = new onEvent(*bbox*, *access_area*)
8: def onEvent(*bbox*, *access_area*)
9: *C_k_* = access_area.getNextCamera()
10: *entry_area* = getAccessAreaFromCameraMetadata(*C_k_*)
11: *bbox* = recognizeByTwoStage(*entry_area*, *C_k_*)
12: return *bbox*

**Table 3 sensors-19-03016-t003:** System Configuration.

Computing Environment:	Workstation
**Processor:**	Intel Core i7 CPU @ 3.20 GHz
**Memory:**	16 GB
**Operating System:**	Windows 10, 64-bit
**Graphics Card:**	Nvidia GTX 1080 GPU

**Table 4 sensors-19-03016-t004:** Camera Configuration.

Camera	Resolution	Frames Per Second(fps)
Cam 1	1920 × 1080	15
Cam 2	1920 × 1080	15
Cam 3	1920 × 1080	15
Cam 4	1920 × 1080	15
Cam 5	1920 × 1080	15
Cam 8	704 × 480	30

**Table 5 sensors-19-03016-t005:** IoU Threshold.

Camera	IoU
Camera 1	94%
Camera 2	90%
Camera 3	95%
Camera 8	81%

**Table 6 sensors-19-03016-t006:** Performance of the proposed method on MOTChallenge database.

Method	MOTA↑	MOTP↑	FAF↓	MT↑	ML↓	FP↓	FN↓	ID sw↓	Frag↓
SORT [39]	33.4	72.1	**1.3%**	11.7%	30.9%	**7318**	32615	1001	1764
TDAM [40]	33.0	**72.8**	1.7%	13.3%	39.1%	10064	**30617**	**464**	1506
MDP [41]	30.3	71.3	1.7%	13.0%	38.4%	9717	32422	680	1500
RMOT [42]	18.6	69.6	2.2%	5.3%	53.3%	12473	36835	684	1282
TC_ODAL [43]	15.1	70.5	2.2%	3.2%	55.8%	12970	38538	637	1716
**STAM-CCF**	**41.2**	72.3	1.97%	**19.5%**	**28.0%**	10815	30775	546	**1040**

The values with bold font are best result than other approaches in Table 6.

**Table 7 sensors-19-03016-t007:** Precision results on CamNeT.

	Number of Gait Sequence				
The First n Frames	20	25	27	30	35
5 frames	0.65	0.65	0.66	0.66	0.65
10 frames	0.64	0.63	0.62	0.62	0.59
15 frames	0.6	0.59	0.59	0.59	0.57

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
