# Peer review of "STAM-CCF: Suspicious Tracking Across Multiple Camera Based on Correlation Filters"

_sensors, 2019, doi:10.3390/s19133016_

Round 1

Reviewer 1 Report

This paper is very interesting. It presents complex solution for people tracking in multiple camera systems. The solution is interesting. But before I will fully accept this paper I ask authors for some improvements. This paper is great for conference but for journal publication it needs more validation and strict description. Main comments are:

1)      The presented system lacks of description of input/output/parameters for each block of calculation (exemplary gait analysis, what are the input data for moment invariants calculation),

2)      There is a lack of qualitative assessment of detection and tracking capability with showing influence of particular steps of processing.

Minor comments:

- Line 231: fi and gi shiuld be Fi anf Gi like in (5).

- Line 275: “two stage gait recognition which never misses”

Reviewer 2 Report

This paper presents a visual tracking method for multiple targets across multiple cameras. 

The method introduces a camera correlation model and re-identification techniques across multiple cameras.

The whole system is somewhat novel. However, the techniques used in the system are not new. YOLO, correlation filter based trackers, IOU scores, and re-identification methods based on the gait sequence are all existing ones. 

There is no comparison and no ablation study for each component in the system. 

Quantitative results are insufficient to demonstrate the effectiveness of the method. 

Round 2

Reviewer 1 Report

The submitted version is improved allowing one to easier follow authors ideas. I agree for publication.

Please only correct one editorial mistake: in title of 3.3.2 section is "Multipoel" and it should be "Multiple"

Reviewer 2 Report

All issues raised by reviewers are carefully addressed in the revised manuscript. 

The experiments are sufficient to demonstrate the effectiveness of the method. 

This manuscript is a resubmission of an earlier submission. The following is a list of the peer review reports and author responses from that submission.